# Whole-genome mapping of APOBEC mutagenesis in metastatic urothelial carcinoma identifies driver hotspot mutations and a novel mutational signature

## Graphical abstract

## Authors

J. Alberto Nakauma-González,
Maud Rijnders, Minouk T.W. Noordsij, ...,
Martijn P.J. Lolkema, Joost L. Boormans,
Harmen J.G. van de Werken

## Correspondence

j.nakaumagonzalez@erasmusmc.nl
(J.A.N.-G.),
h.vandewerken@erasmusmc.nl
(H.J.G.v.d.W.)

## In brief

Nakauma-González et al. analyzed the mutations associated with APOBEC enzymes in whole genomes of bladder and other cancers. They discovered pairs of recurrent mutations in DNA prone to forming hairpin-loop structures, named didymi. This and other genomic features were modeled to identify drivers of cancer associated with APOBEC across 3,859 samples.

## Highlights

- APOBEC enzymes mutate cancer genomes and affect >90% of urothelial cancers

- APOBEC mutates recurrent genomic positions, creating thousands of hotspot mutations

- Didymi are hotspot mutations in hairpin loops with a specific mutational signature

- Analysis of 3,859 cancer whole genomes identified 31 APOBEC hotspots as cancer drivers

Nakauma-González et al., 2024, Cell Genomics 4, 100528
April 10, 2024 © 2024 The Authors.

CellPress

# Whole-genome mapping of APOBEC mutagenesis in metastatic urothelial carcinoma identifies driver hotspot mutations and a novel mutational signature

J. Alberto Nakauma-González,[1,2,3,*] Maud Rijnders,[3] Minouk T.W. Noordsij,[1] John W.M. Martens,[3] Astrid A.M. van der Veldt,[3,4] Martijn P.J. Lolkema,[3,6] Joost L. Boormans,[2] and Harmen J.G. van de Werken[1,2,5,7,*]

[1]Cancer Computational Biology Center, Erasmus MC Cancer Institute, University Medical Center Rotterdam, 3015 GD Rotterdam, the Netherlands
[2]Department of Urology, Erasmus MC Cancer Institute, University Medical Center Rotterdam, 3015 GD Rotterdam, the Netherlands
[3]Department of Medical Oncology, Erasmus MC Cancer Institute, University Medical Center Rotterdam, 3015 GD Rotterdam, the Netherlands
[4]Department of Radiology & Nuclear Medicine, Erasmus MC, University Medical Center Rotterdam, 3015 GD Rotterdam, the Netherlands
[5]Department of Immunology, Erasmus MC Cancer Institute, University Medical Center Rotterdam, 3015 GD Rotterdam, the Netherlands
[6]Present address: Amgen Europe BV, 4817 ZK Breda, the Netherlands
[7]Lead contact
*Correspondence: j.nakaumagonzalez@erasmusmc.nl (J.A.N.-G.), h.vandewerken@erasmusmc.nl (H.J.G.v.d.W.)

## SUMMARY

Apolipoprotein B mRNA-editing enzyme catalytic polypeptide-like (APOBEC) enzymes mutate specific DNA sequences and hairpin-loop structures, challenging the distinction between passenger and driver hotspot mutations. Here, we characterized 115 whole genomes of metastatic urothelial carcinoma (mUC) to identify APOBEC mutagenic hotspot drivers. APOBEC-associated mutations were detected in 92% of mUCs and were equally distributed across the genome, while APOBEC hotspot mutations (ApoHMs) were enriched in open chromatin. Hairpin loops were frequent targets of didymi (twins in Greek), two hotspot mutations characterized by the APOBEC SBS2 signature, in conjunction with an uncharacterized mutational context (Ap[C>T]). Next, we developed a statistical framework that identified ApoHMs as drivers in coding and non-coding genomic regions of mUCs. Our results and statistical framework were validated in independent cohorts of 23 non-metastatic UCs and 3,744 samples of 17 metastatic cancers, identifying cancer-type-specific drivers. Our study highlights the role of APOBEC in cancer development and may contribute to developing novel targeted therapy options for APOBEC-driven cancers.

## INTRODUCTION

Cancer genomes accumulate somatic mutations via different mutagenic processes, and one of the most common is attributed to the apolipoprotein B mRNA-editing enzyme catalytic polypeptide-like (APOBEC) family.[1] APOBEC has a specific mutational signature, which is characterized by C>T/G mutations in the TpC context and is captured in the SBS2 and SBS13 signatures, as defined by the Catalog of Somatic Mutations in Cancer (COSMIC).[2] In some tumor types with high APOBEC activity, the contribution to the tumor mutational burden is substantial, which increases the neoantigen load, favoring response to immune checkpoint inhibitors.[3,4] However, APOBEC is also responsible for the emergence of driver mutations that contribute to cancer development, as shown in mouse models.[5] Discriminating driver events from passenger events is essential to reconstruct the evolutionary history of cancers and identify effective novel drug targets in APOBEC-driven tumors.

The mutational process of APOBEC has been extensively studied, revealing its preference for single-stranded DNA structures that form hairpin loops.[6] This characteristic of APOBEC could result in identical somatic mutations in tumors from multiple patients, so-called hotspot mutations or hotspots. Due to their high prevalence, these hotspots can be assigned erroneously as driver mutations, especially in the non-coding area of the genome. However, the vast majority of mutations are passengers and do not contribute to cancer development, and the same principle may also apply to hotspot mutations.[7] Although bioinformatics strategies to identify driver hotspot mutations have been developed,[8,9] the unique characteristics of the APOBEC mutagenic process require specific considerations to accurately account for all co-variables.

APOBEC-derived mutations are a dominant contributor to the mutational landscape in urothelial carcinoma (UC). Therefore, we analyzed whole-genome DNA sequencing data of 115 metastatic UC (mUC) and matched blood samples[10] to identify driver hotspot mutations in the context of APOBEC mutagenesis. The comprehensive characterization of APOBEC-enriched tumors identified a novel mutational signature associated with DNA mismatch repair as well as genomic co-variates associated



with APOBEC-derived hotspot mutations (ApoHMs), which we used to develop a statistical framework and identify driver ApoHMs. Furthermore, our findings were validated in whole genomes of an independent cohort of 23 non-mUCs, and the analysis was extended to include 442 metastatic breast cancer (mBC) and 3,302 samples of 16 other metastatic cancer types.

## RESULTS

### APOBEC mutagenesis dominates the mutational landscape of UC

The analysis of whole-genome sequencing (WGS) data of mUCs and matched blood samples revealed a median of 20,667 (Q1 [quartile 1] = 14,304, Q3 [quartile 3] = 31,411) single-nucleotide variants (SNVs) per tumor. mUCs with a significant enrichment (E) for C>T mutations in the TCW (W = A or T) context were considered APOBEC positive (92%). These tumors were further stratified according to APOBEC enrichment as APOBEC high (41%, E > 3), APOBEC medium (33%, 2 < E < 3), and APOBEC low (18%, E > 1) (Figure 1A). The median contribution of APOBEC COSMIC signatures (SBS2+SBS13) in APOBEC-high, -medium, and -low tumors was 61%, 37%, and 15%, respectively. For the remaining 8% of tumors lacking APOBEC mutations, the median APOBEC signature was less than 2%, potentially reflecting the noise of the mutational signature calling. We associated the APOBEC stratification with multiple factors to better assess the different APOBEC subtypes. Tumor purity, for instance, declined with increasing APOBEC mutagenesis (Figure S1). Moreover, age was associated with the enrichment of APOBEC mutations (Figure S1). The median clonal fraction of SNVs was lower in tumors with APOBEC mutations than in non-APOBEC tumors, suggesting higher tumor heterogeneity in APOBEC-enriched tumors (Figure S1). Localized hypermutation events (kataegis) strongly correlated with APOBEC enrichment (Spearman r = 0.80, p < 0.001). Homologous recombination (HR) deficiency (n = 3) was only present in APOBEC-low tumors (Fisher's exact test, p = 0.005), while none of the patients with microsatellite instability (MSI; n = 4) had evidence of APOBEC mutagenesis (Fisher's exact test, p < 0.001). Structural variants were more frequent in APOBEC tumors than in non-APOBEC tumors (Figure S1). Additionally, APOBEC tumors had a higher ploidy (median ploidy = 3) and a higher number of genes affected by copy number alterations (CNAs) than non-APOBEC tumors (Figure S1), suggesting genomic instability in APOBEC-driven mUC tumors. APOBEC mutagenesis was not associated with sex, the primary origin of mUC (upper tract versus bladder), or chromothripsis.

### APOBEC mutagenesis is an ongoing process in metastatic lesions of UC

Next, we analyzed RNA sequencing data of 90 matched samples of mUC. Pathway activity based on downstream gene expression, such as cell cycle or p53, was similar between the APOBEC groups (Figure S2). Similarly, analysis of APOBEC expression of all genes of the APOBEC family (APOBEC1 was not expressed) revealed no significant differences between APOBEC and non-APOBEC tumors (Figure 1B). We detected a weak positive correlation between the expression of APOBEC3A and APOBEC3B (Figure 1C). To further investigate the mutagenic activity of both

enzymes, the fold enrichment of C>T and C>G mutations, at the DNA level, in the tetrabase YTCA (related to APOBEC3A; Y are pyrimidine bases) and RTCA (related to APOBEC3B; R are purine bases) context was calculated[11] (Figure 1D). In both cases, YTCA and RTCA mutations did not correlate with expression of APOBEC3A or APOBEC3B (Figure S3). The lack of correlation might be linked to the heterogeneous expression of APOBEC enzymes that oscillate throughout the cell cycle.[12,13] Furthermore, we detected that both APOBEC3A and APOBEC3B contributed to APOBEC-associated mutations (fold enrichment above 1.0). Nevertheless, APOBEC3A appeared to be the main contributor, as suggested in primary cancers.[11,14] Considering the mRNA expression of both APOBEC3A and APOBEC3B enzymes, we calculated the APOBEC expression score (the sum of the normalized expression of APOBEC3A and APOBEC3B). It appeared that the level of APOBEC enrichment correlated with the APOBEC expression score (Figure 1E). This analysis confirmed the link between APOBEC RNA expression at the time of biopsy and the historical accumulation of APOBEC-associated mutations in mUC that others have reported in primary UC.[15,16]

Recently, it was proposed that edited DDOST mRNA can be used to measure ongoing APOBEC mutagenesis.[17] We found that the frequency of C>U alterations in the DDOST mRNA at position chr1:20981977 was enriched in tumors with APOBEC mutagenesis, with up to 15% of mRNA reads edited in one single sample (Figure 1F). Additionally, we analyzed ongoing APOBEC mutagenesis in mUC by WGS of eight tumors from patients who had undergone serial biopsies of metastatic lesions (Figure S4). We observed that the APOBEC mutational signature was present in private mutations of the second biopsy of these patients, suggesting that APOBEC mutagenesis could be active in the period between the first and second biopsy (Figure S4A). A lower cancer cell fraction (normalized allele frequency by copy number and purity; STAR Methods) in private SNVs of the second biopsy compared with shared (trunk) mutations confirms that these mutations were acquired more recently, as they are only present in a subpopulation of cancer cells (Figure S4B). This result, together with the APOBEC signature detected in the subclonal mutations (Figure S5), suggests that the presence of subclonal populations due to APOBEC mutagenesis may contribute to the ongoing evolution of UC in the metastatic setting.

### ApoHMs are enriched in highly accessible genomic regions

The high resolution achieved by WGS allowed us to investigate the enrichment of APOBEC and non-APOBEC mutations (non-TpC context) in specific genomic regions. We found that the number of non-APOBEC-associated SNVs, for instance, varied across the genome (Figure 2A). When this distribution overlapped with DNA accessibility and overall gene expression level, the frequency of non-APOBEC mutations decreased in open chromatin (highly accessible regions) and highly transcribed regions (Figure 2B). In contrast, the frequency of APOBEC-associated mutations was nearly constant across the genome. When restricting the analysis to APOBEC tetrabase mutations, we found that the difference in the distribution of Y/RTCA mutations across genomic regions decreases with the level of APOBEC enrichment (Figure 2C). Interestingly, in APOBEC-high tumors, YTCA mutations were evenly

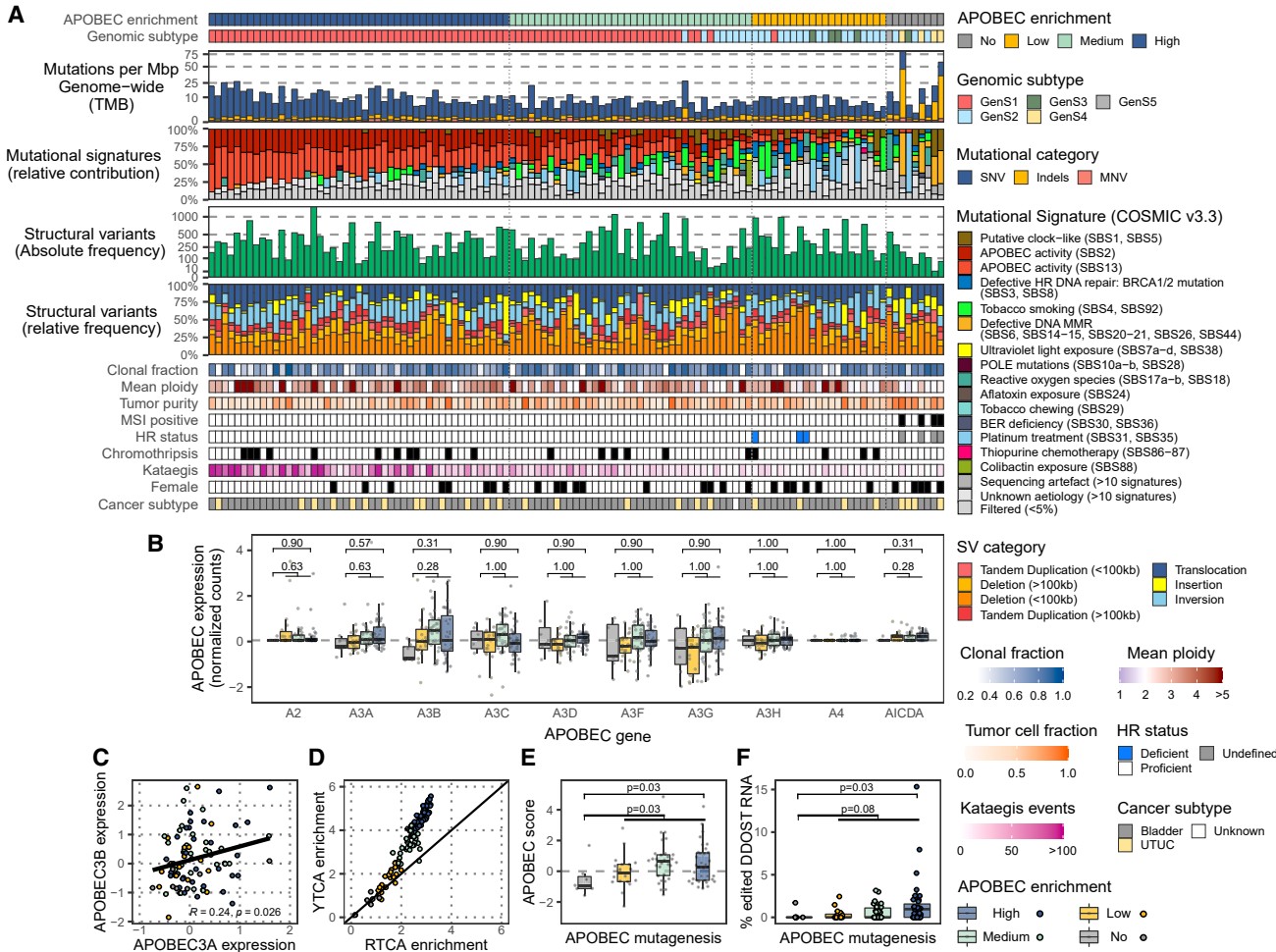

**Figure 1. Genomic landscape and APOBEC activity of mUC (n = 115) stratified by APOBEC enrichment**

(A) WGS data of metastatic urothelial carcinoma (mUC) were classified according to the enrichment of APOBEC-associated mutations as having high, medium, low, or no APOBEC enrichment. The genomic features are displayed from top to bottom as follows: APOBEC mutagenesis; genomic subtype (GenS1–GenS5) as described previously;[10] genome-wide tumor mutational burden (TMB); mutational signatures grouped by etiology, with both APOBEC signatures shown separately; absolute frequency of structural variants (SVs); relative frequency of SV categories; clonal fraction; ploidy; tumor purity; microsatellite instability (MSI) status; homologous recombination (HR) deficiency status; samples with at least one chromothripsis event; frequency of kataegis events; female patients; and primary origin of mUC (upper tract versus bladder).

(B) Expression of *APOBEC* and *AICDA* genes in 90 samples with available RNA sequencing data.

(C) Pearson correlation of RNA expression of *APOBEC3A* and *APOBEC3B*.

(D) Fold enrichment of C>T and C>G alterations in the YTCA (related to APOBEC3A; Y are pyrimidine bases) and RTCA (related to APOBEC3B; R are purine bases) context.

(E) APOBEC score (normalized expression of *APOBEC3A* + *APOBEC3B*).

(F) Percentage of mRNA C>U mutations in *DDOST* at position chr1:20981977.

In (B), (E), and (F), the Wilcoxon rank-sum test was applied to compare APOBEC tumors vs. non-APOBEC tumors. p values were Benjamini-Hochberg corrected in (B). Box plots show the median, inter-quartile range (IQR: Q1–Q3) and whiskers (1.5 × IQR from Q3 to the largest value within this range or 1.5 × IQR from Q1 to the lowest value within this range). See also Figures S1–S5.

distributed. In contrast, RTCA mutations were enriched in low-DNA-accessible and low-transcribed regions, although this enrichment was considerably less compared with tumors with lower levels of APOBEC mutations.

Because of the high correlation between kataegis and APOBEC enrichment, we analyzed the distribution of kataegis loci across the genome. Contrary to the overall distribution of SNVs, our data suggest that kataegis events are more likely to occur in re-

gions with high DNA accessibility and high transcriptional activity (Figure 2B). Moreover, we also evaluated the genome-wide distribution of all hotspot mutations (two or more mutations in a specific genomic position), representing 0.35% of all mutated genomic positions. We found that the frequency of highly recurrent (n ≥ 4) ApoHMs were enriched in high-DNA-accessibility and highly transcriptionally active regions (Figure 2D). Thus, while general APOBEC mutagenesis seemed to occur uniformly across the

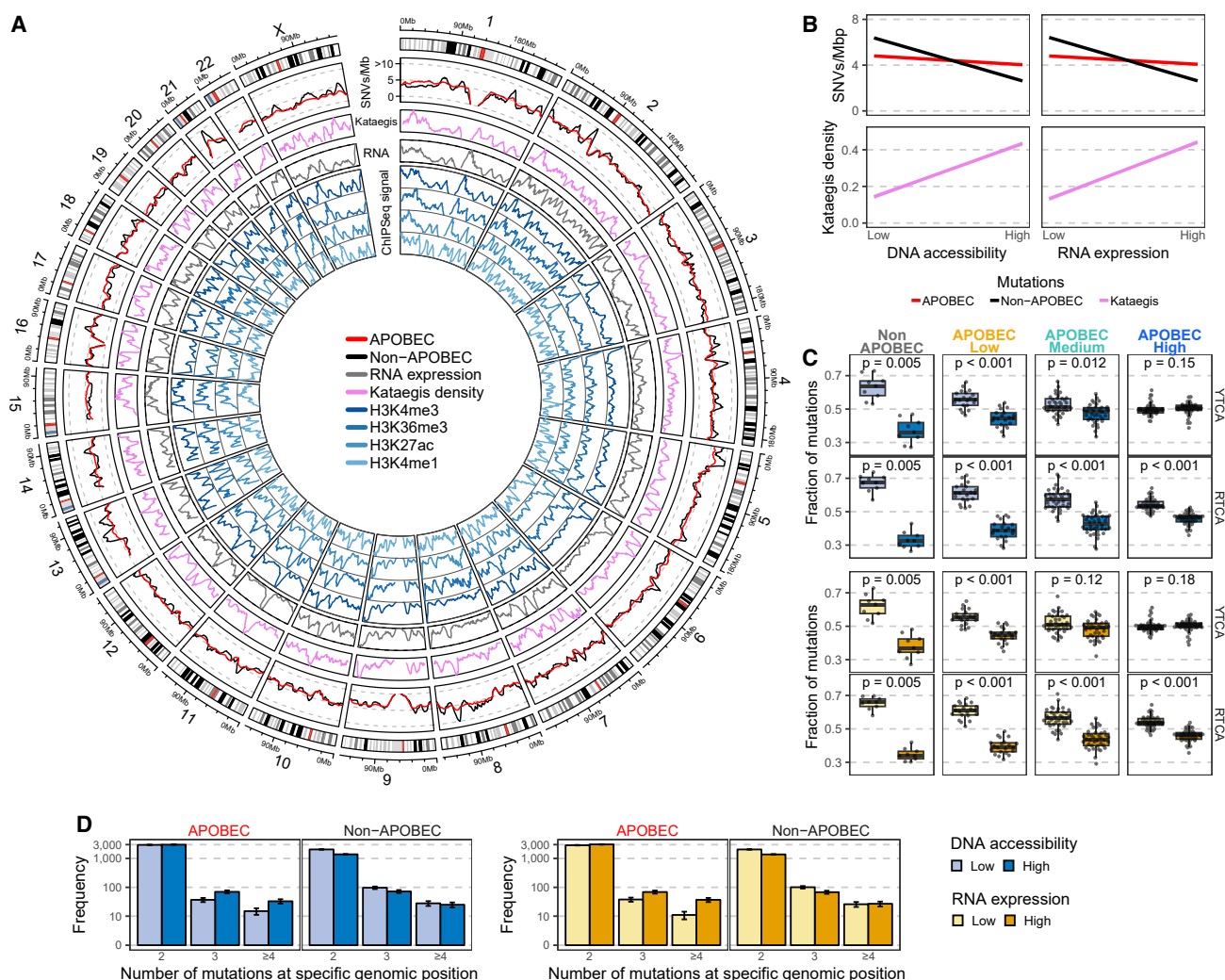

**Figure 2. Distribution of APOBEC-associated mutations across genomic regions of mUC**

(A) Whole-genome sequencing data (n = 115) were analyzed to estimate the mean number of single-nucleotide variants (SNVs) in windows of 1 mega base-pair (Mbp) across the entire genome. The circos plot shows, from outer to inner circles, the genomics ideogram from chromosome 1 to X, where the centrosomes are indicated in red; the mutational load of APOBEC- and non-APOBEC-associated mutations (mutations in the TpC or non-TpC context, respectively); the density of kataegis events; average RNA counts (expression) from tumors with available RNA sequencing data (n = 90); and DNA accessibility estimation from different chromatin immunoprecipitation sequencing (ChIP-seq) experiments of multiple histone marks from normal urothelial samples derived from the Encyclopedia of DNA elements (ENCODE).[18] Peaks represent highly accessible DNA.

(B) Linear regression of the mutational load for APOBEC- and non-APOBEC-associated mutations as well as the density of kataegis events across the genome with DNA accessibility and expression data.

(C) Relative distribution of APOBEC YTCA (related to APOBEC3A; Y are pyrimidine bases) and RTCA (related to APOBEC3B; R are purine bases) mutations across DNA-accessible and RNA expression regions. Samples were stratified according to the level of APOBEC mutagenesis. The Wilcoxon signed-rank test was applied, and p values were Benjamini-Hochberg corrected. Box plots show the median, inter-quartile range (IQR: Q1–Q3) and whiskers (1.5 × IQR from Q3 to the largest value within this range or 1.5 × IQR from Q1 to the lowest value within this range).

(D) Frequency of hotspot mutations grouped according to APOBEC- and non-APOBEC-associated mutations and DNA accessibility or RNA expression level. Error bars represent the mean ± standard error.

---

genome, kataegis and ApoHMs seemed to occur more frequently in open chromatin and highly transcribed loci.

## Recurrent hotspot mutations correlate with APOBEC mutagenesis

Next, we investigated the genomic consequence of hotspot mutations and found that the most frequent hotspot mutations

in mUC occurred in non-coding regions of the genome (Figure 3A). Hotspot mutations in the *TERT* promoter were present in 62% of tumors. In line with previous reports,[19,20] *TERT* expression did not differ between tumors with hotspot mutations and those being wild type (Figure S6A). However, differential gene expression analysis showed that tumors with hotspot mutations in the *TERT* promoter had high

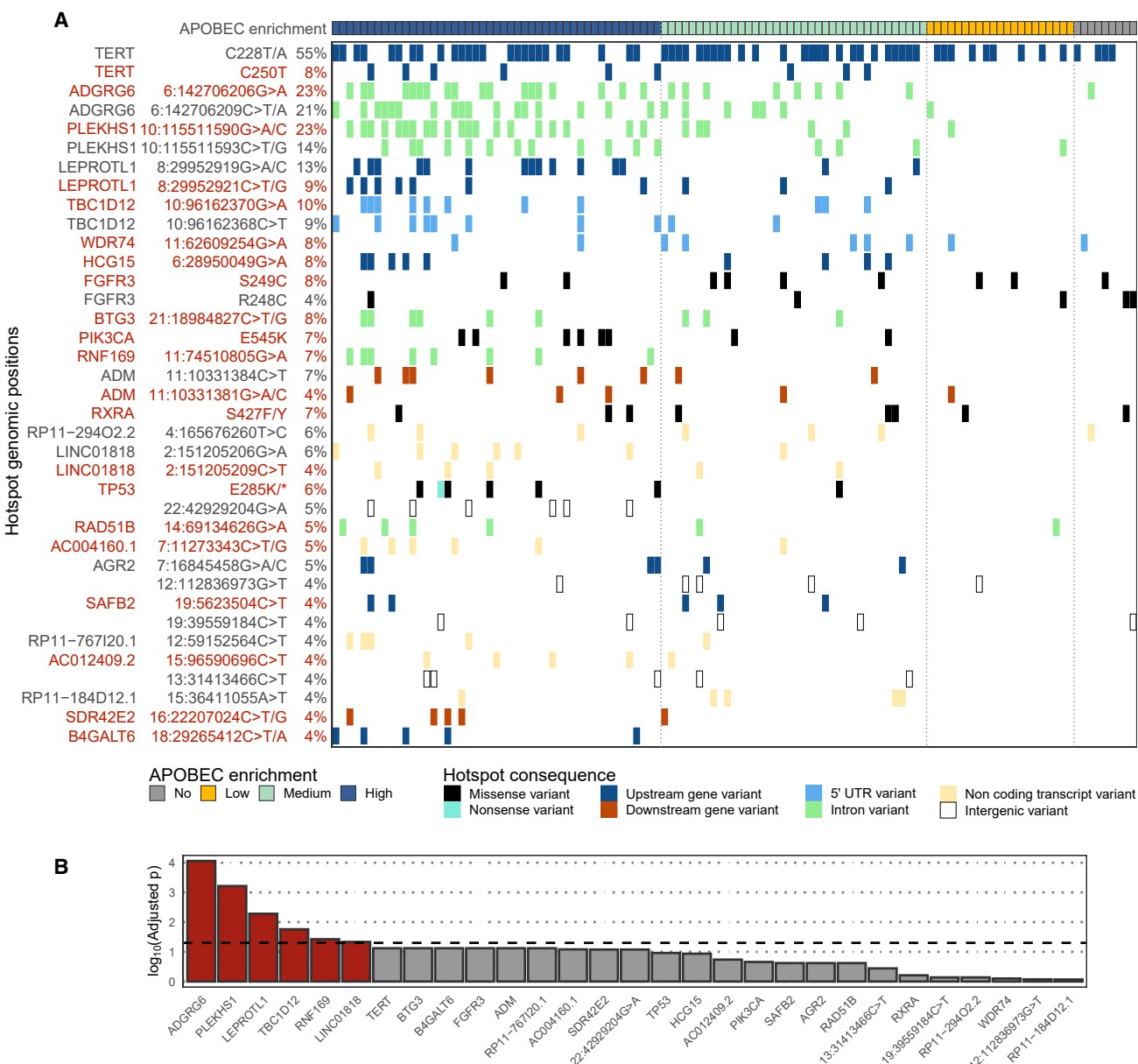

**Figure 3. Recurrent hotspot mutations of mUC correlate with APOBEC mutagenesis**
(A) Overview of recurrent hotspot mutations present in at least five samples, including all substitutions occurring in the same genomic position. Hotspot mutations occurring in the TpC context are highlighted in red.
(B) The association of hotspot mutations and APOBEC fold enrichment (continuous values) was interrogated with a logistic regression analysis applying the Wald test. p values were corrected using the Benjamini-Hochberg method and ordered accordingly. Bars above the dashed line (−log10(0.05)) are statistically significant and indicated in red. See also Figure S6 and Table S1.

expression of genes related to the biological oxidation pathway (Figure S6B; Table S1). Besides *TERT*, other frequent hotspot mutations were identified in the non-coding regions near *ADGRG6* (40%), *PLEKHS1* (33%), *LEPROTL1* (20%), and *TBC1D12* (15%). Similarly, these hotspot mutations did not affect the expression of these genes but were associated with transcriptomic effects in several genes (Figure S6) and biological pathways (Table S1). These hotspot mutations

strongly correlated with enrichment for APOBEC-associated mutations, suggesting their origin in APOBEC mutagenesis (Figure 3B).

All frequent hotspot mutations in the coding region have been described previously and affected known driver genes: *FGFR3* S249C/R248C (8% and 4%), *PIK3CA* E54K (7%), *RXRA* S427F/Y (7%) and *TP53* E285K/* (6%). Comparing the expression of these known driver genes affected by hotspot mutations

vs. the wild type, only *FGFR3* hotspot mutations significantly affected the expression of this gene (Figure S6A).

### Hairpin loops are targets of twin hotspot mutations called didymi

We noticed that the hotspot mutations near *ADGRG6*, *PLEKHS1*, *LEPROTL1*, and *TBC1D12* are located within DNA hairpin-loop structures. It is known that DNA hairpin loops are targets of APOBEC3A (Figure 4A);[21,22] therefore, we predicted DNA hairpin loops for all mutated genomic positions (STAR Methods). The predicted hairpin loops near *ADGRG6*, *PLEKHS1*, *LEPROTL1*, and *TBC1D12* are each affected by two hotspot mutations, which are referred to as twin mutations.[22] Moreover, we noticed that the twin mutations were not mutually exclusive, which differs from the hotspot mutations in the *TERT* promoter (mutual exclusivity test, p < 0.001). Only the twin hotspot mutations in *TBC1D12* co-occurred more frequently than expected among APOBEC-high tumors (p = 0.02). Further analysis of co-occurred twin hotspot mutations revealed that very few had identical variant allele frequencies, suggesting that most twin mutations in the same tumor occurred in independent events, as they were also found on different alleles (Figure S7).

Next, we investigated the properties and origin of these twin mutations. A comprehensive analysis of all DNA hairpin-loop structures in the human genome affected by two mutations in their loops revealed 2,387 twin mutations (4,774 altered genomic positions), representing 0.16% of all mutated genomic positions. The distance between twin mutations varied, but when the frequency of mutations increased, the distance decreased to mainly one or two bases and the loop sizes to mainly three to four bases (Figure 4B). Additional examination of the 96 tri-nucleotide contexts of all twin mutations revealed that both APOBEC COSMIC signatures, SBS2 and SBS13, were dominant (Figure 4C). However, at higher mutational frequencies (n ≥ 5), only signature SBS2 remained. We also observed a secondary signature of C>T mutations in the ApC context that does not resemble any known COSMIC signature (Figure S8, and Table S2). The absolute contribution of this signature was similar across all APOBEC tumors, and its prevalence in the mUC cohort correlated with spontaneous deamination (SBS1) and defective DNA mismatch repair signatures (Figure S9; SBS6, SBS15, SBS20, and SBS26).

Furthermore, APOBEC-driven tumors were enriched for twin mutations occurring only in the TpC context (Figure 4D). Contrary to the general pattern of ApoHMs (enriched in DNA-accessible regions), twin mutations with a high number of alterations were similarly distributed between high- and low-DNA-accessible regions (Figure 4E).

Additionally, we found that, at higher mutational frequency, at least one of the twin mutations occurs in the TpC context (Figure 4F) and that the sequence between the two is 1001 or 101 (0 = A/T, 1 = G/C; underlined are the positions of the twin mutations) (Figure 4G). Because of the unique characteristics of frequently affected twin mutations, we named them didymi (twins in Greek). In summary, didymi are two C>T hotspot mutations found in DNA hairpin loops separated by one or two A/T base pairs in which at least one of the twin mutations is located in the TpC context and the other in

NpC (N = any base pair; most N bases are A or T). Applying this definition, we identified 231 didymi in the mUC cohort (Table S3).

### Driver hotspot mutations in UC

After identifying several hotspot mutations that could be attributed to APOBEC activity, we assessed whether these hotspot mutations had a selective advantage (drivers) or not (passengers). Recent attempts relying on the stability of hairpin loops have been proposed to differentiate passengers from driver ApoHMs.[8,21] We confirmed that a more stable loop (Gibbs free energy ΔG; STAR Methods) leads to a higher number of alterations (Figure 5A). Taking this into account, we developed a statistical model to identify driver hotspot mutations, considering not only the stability of hairpin loops but also the tri-nucleotide context, DNA accessibility, and the potential for didymi via sequences in the loop (Figure 5A).

Putative ApoHMs were divided into those located outside or inside the loop of DNA hairpin-loop structures. In the case of ApoHMs outside of hairpin loops, only those in the TpC context were considered. For ApoHMs within loops, all alterations in the TpC, ApC, CpC, and GpC context were included in the analysis to account for didymi. In the case of TpC, the distribution of hotspot mutations in the trinucleotide context was considered. A background distribution was modeled as a Poisson process, and the significant enrichment of mutations in a particular genomic site was estimated.

We identified 0.40% (n = 27) of ApoHMs as drivers (adjusted p < 0.05; Figure 5B). Known driver genes affected by hotspot mutations included coding alterations in *TP53*, *PIK3CA*, *FGFR3*, *RXRA*, and the *TERT* promoter. All other putative driver ApoHMs affected non-coding regions, including didymi in *ADGRG6*, *PLEKHS1*, *TBC1D12*, and *LEPROTL1*, proposed as drivers by other studies.[9,23] Other potential driver ApoHMs include *RNF169* (involved in DNA damage repair),[24] *BTG3* (angiogenesis),[25] *ADM* (adrenomedullin, a vasodilator),[26] *GDF3* (regulation of transforming growth factor β [TGF-β]),[27] and WDR74 (ribosome biogenesis).[28]

To validate our method and confirm the driver ApoHM assessment, we used an independent cohort of non-mUC of the bladder (n = 23) of the Pan-cancer Analysis of Whole Genomes (PCAWG) study[29] (Figure S10). This analysis confirmed the previously identified ApoHMs as potential cancer drivers of UC. Moreover, in this cohort, 96% of tumors were APOBEC driven, APOBEC enrichment correlated with kataegis, and the largest group (35%) had high enrichment for APOBEC-associated mutations.

Furthermore, the performance of the model to identify driver ApoHMs in hairpin loops was tested. The quantile-quantile (Q-Q) plots show that the empirical distribution of ApoHMs deviates from the theoretically expected distribution (Kolmogorov-Smirnovtest, p < 0.001). However, when outliers that represent highly frequent ApoHMs (>10) are excluded, which, according to our analysis, are all drivers, we observed a good agreement between our model and the theoretical distribution (Figure S11A; Kolmogorov-Smirnovtest, p = 0.19). By simulating a synthetic genome of mUCs, we showed that an 80% statistical power is reached when the cohort size is ~75 samples for highly

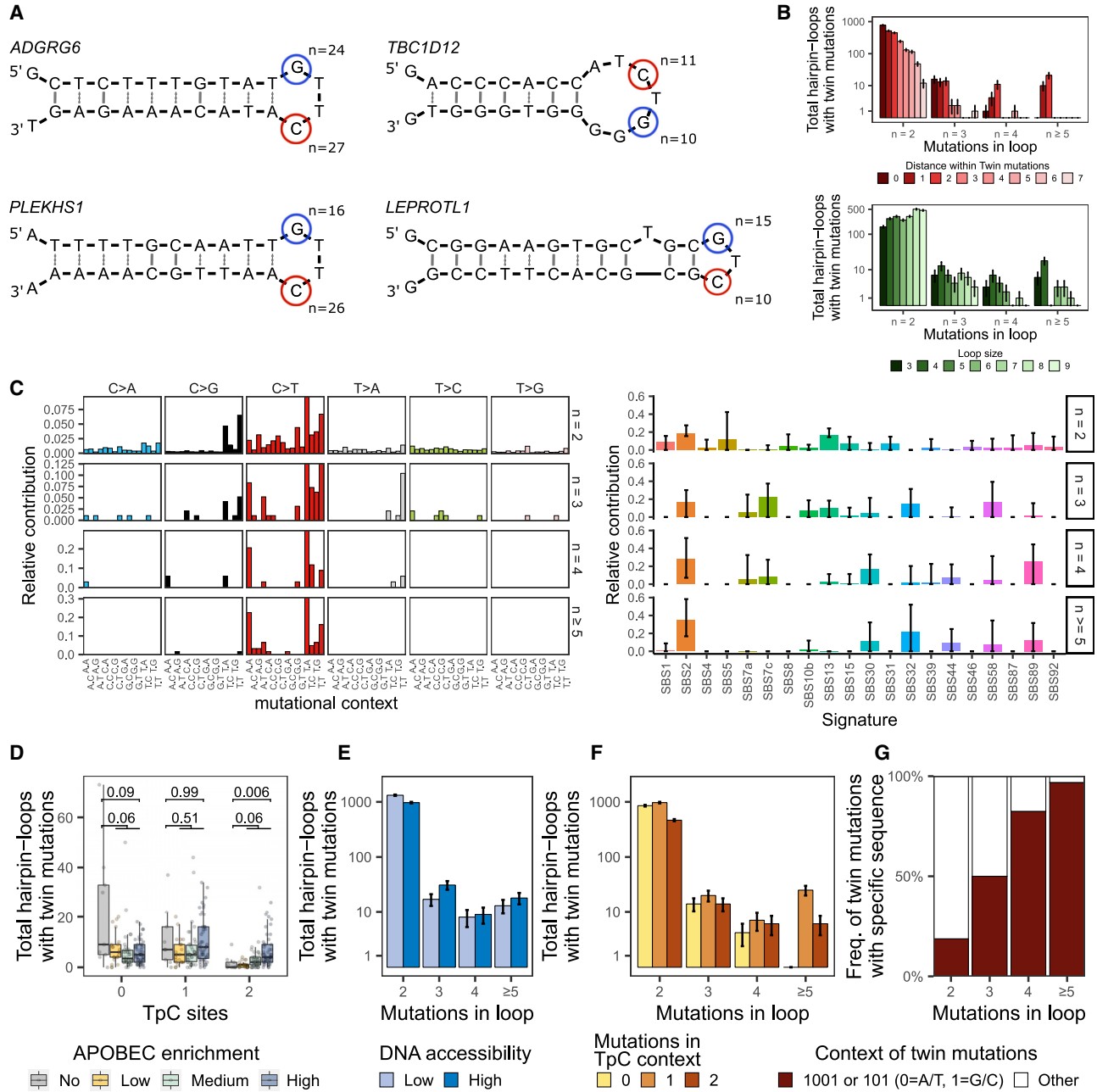

**Figure 4. Genomic characteristics of twin hotspot mutations in mUC**

(A) Hairpin-loop structures affected by frequent hotspot mutations in *ADGRG6*, *TBC1D12*, *PLEKHS1*, and *LEPROTL1*. The positions of hotspot mutations are marked in red for TpC context and blue otherwise.

(B) Distribution of twin mutations according to the distance between twin mutations and loop size. Error bars represent the mean ± standard error.

(C) Mutational signatures (COSMIC v.3.3) of twin mutations according to their frequency. The stability of the signature call was tested by applying 1,000 bootstrap iterations (error bars represent the mean ± 1.96 standard deviations). Only SBS2 was very stable in highly frequent (n ≥ 5) twin mutations.

(D) Frequency distribution of hairpin loops affected by twin mutations according to APOBEC mutagenesis. The Wilcoxon rank-sum test was applied to compare APOBEC vs. non-APOBEC tumors, and p values were Benjamini-Hochberg corrected. Box plots show the median, inter-quartile range (IQR: Q1–Q3) and whiskers (1.5 × IQR from Q3 to the largest value within this range or 1.5 × IQR from Q1 to the lowest value within this range).

(E) DNA accessibility, (F) Number of mutations in the TpC context within a loop and (G) DNA sequence between twin mutations. Error bars represent the mean ± standard error. See also Figures S7–S9 and Tables S2 and S3.

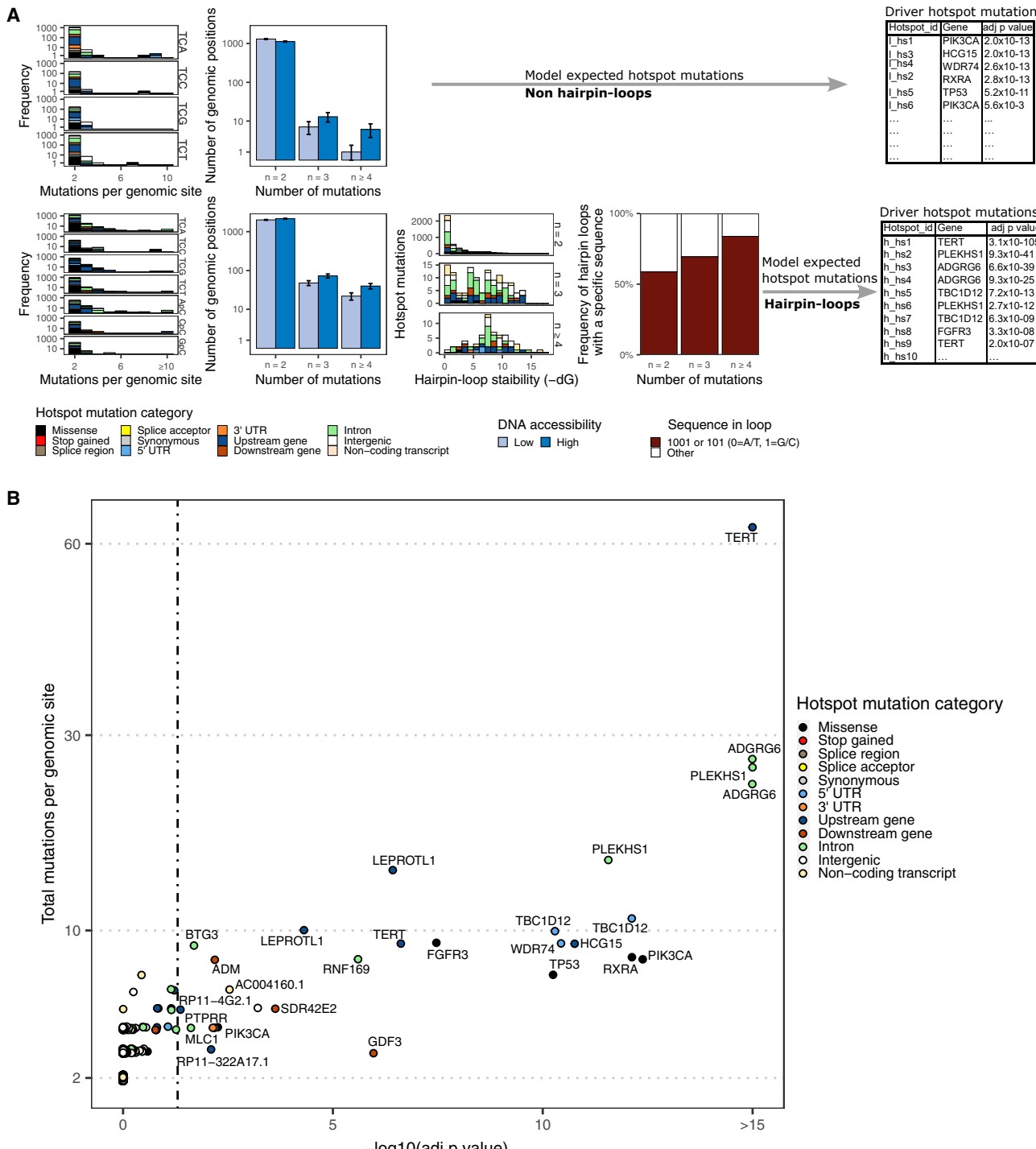

**Figure 5. Driver hotspot mutations associated with APOBEC mutagenesis in UC**

(A) Frequency distribution of variables (trinucleotide context, DNA accessibility, DNA hairpin-loop stability, and sequence in the loop) considered to identify hotspot mutations that were more frequently mutated than expected outside (top) and within (bottom) hairpin-loop structures. Error bars represent the mean ± standard error.

(B) Driver hotspot mutations were estimated separately for mutations outside and within hairpin loops. Per group, p values were corrected using the Benjamini-Hochberg method (adj p value). See also Figures S10 and S11 and Table S4.

frequent (>10%) driver ApoHMs (Figure S11B). However, the power to detect rare driver ApoHMs (≤10%) is considerably reduced, and a larger cohort is needed. We also evaluated the contribution of different genomic features as covariates (Table S4) to identify driver ApoHMs. The McFadden's $R^2$ in the model that only considers the trinucleotide context is low ($R^2 = 0.04$), but the goodness of fit increases when considering the hairpin loop ($R^2 = 0.23$) and hairpin loop + sequence in the loop ($R^2 = 0.27$) and when adding DNA accessibility regions ($R^2 = 0.28$) into the model. DNA accessibility shows a high (anti-)correlation with other genomic features: GC content, RNA expression levels, mutational load, methylation, and replication timing (Figure S11C). Therefore, the addition of the other variables in the model has limited added value (Figure S11D).

### Driver hotspot mutations in mBC

To test our statistical framework in other cancer types and to evaluate whether our findings were UC specific, we analyzed a cohort of 442 mBCs (Figure 6).[30] Similar to UC, breast cancer is commonly affected by APOBEC mutagenic activity.[1] We identified APOBEC-enriched tumors in 76% of patients, and only 19% of mBC tumors were APOBEC high. In most patients (39%), tumors were classified as APOBEC low and were enriched for HR deficiency (two-sided Fisher's exact test, p < 0.001).

The most frequent coding hotspot mutations affected *PIK3CA*, *ESR1*, and *AKT1*. Twin mutations in hairpin loops displayed a similar mutational signature as those in mUC, including the APOBEC signature SBS2 in conjunction with the uncharacterized C>T mutations in the ApC context that define didymi (Figures S12A and S12B). Didymi in *PLEKHS1* and *ADGRG6* were the most frequent non-coding hotspot mutations, while *LEPROTL1* and *TBC1D12*, two other didymi frequently found in mUC, only affected 1% or less of mBCs. A total of 694 didymi were identified in mBCs (Table S5), but only 19 (2.7%) were shared with mUC (Figure S12C).

Our analysis revealed 51 driver ApoHM in APOBEC-enriched mBCs (Figure 6B), representing only 0.07% of all ApoHMs. Drivers included missense hotspot mutations in *PIK3CA*, *AKT1*, and *TP53* and hotspot mutations outside of the protein-coding region of *MAPKAPK2* and *STAG1* and including didymi in *PLEKHS1* and *ADGRG6*. In contrast to mUC, and despite being one of the most frequently affected genes by hotspot mutations in mUC, *LEPROTL1* was not a driver of mBC. This analysis suggests that driver mutations derived from APOBEC activity are cancer type specific.

### Driver hotspot mutations across multiple metastatic cancers

APOBEC mutagenic activity is widespread across multiple cancer types. Here, we analyzed the genome of 16 additional metastatic cancers, which, in total, represents 3,302 whole genomes (+ 115 mUCs + 442 mBCs = 3,859). Urothelial, breast, and uterine cancers have the highest proportion of APOBEC-high tumors (Figure 7A). In mUC and mBC, 95% of hotspot mutations affect non-coding transcripts, introns, or intergenic regions. This proportion varies per cancer type and can represent up to 99% of all hotspot mutations in esophageal cancer. Missense hotspot

mutations are rare, but the highest proportion is found in liver cancer, reaching 5%.

The frequency of ApoHMs increases with the strength of APOBEC mutagenesis, and they are more recurrent in hairpin loops (Figure 7B). However, skin cancer does not follow this pattern, which has been proven to be problematic in other studies due to its hypermutated nature, inflating the number of driver events.[31,32] These studies make special considerations or exclude skin cancer altogether from their analysis. We found that skin cancer is mostly defined as an APOBEC-low cancer type, and we suspected that a large proportion of mutations that have the APOBEC signature may not have been derived from APOBEC mutagenic activity. This is supported by the relatively low number of driver ApoHMs in skin cancer that correlates with APOBEC enrichment (Figure 7C), despite many ApoHMs defined as drivers by our model (Table S6). These driver ApoHMs, considered "true" APOBEC-derived mutations, are more frequent in breast, urothelial, lung, and uterine cancers. *TERT*, *PIK3CA*, *PLEKHS1*, and *ADGRG6* are the most affected genes by driver ApoHMs. All four genes harbor two driver ApoHMs, which are targeted by APOBEC, except for *TERT*, which has only one "true" ApoHM (C250T). *TP53* is another gene that is frequently affected by driver ApoHMs; however, only one of nine is a "true" APOBEC-derived mutation. The pan-cancer overview exhibits the distribution of driver ApoHMs in APOBEC-enriched cancers, and the power gained by integrating 3,859 samples revealed that, of all potential drivers, only 31 might be "true" APOBEC-derived mutations.

## DISCUSSION

In this study, we describe the genomic landscape of APOBEC-driven tumors, characterize ApoHMs, and identify potential cancer drivers in mUC. The in-depth analysis of 115 whole genomes of mUC identified chromatin accessibility, hairpin loop stability, and specific sequences within the hairpin loop as variables associated with ApoHMs. These variables, in combination with the mutational context, were used to identify ApoHMs that were more frequently mutated than expected by chance.

The substrate of APOBEC enzymes is single-stranded DNA (ssDNA),[1] which has led to the following conflicting hypotheses: (1) APOBEC enzymes are mainly active during replication,[33] and (2) APOBEC is mainly active in open chromatin and transcriptionally active genomic regions.[34] The equal distribution of all APOBEC-associated mutations across genomic regions supports the hypothesis that these mutations had been generated during replication, when APOBEC enzymes have equal access to ssDNA across the genome.[33] However, kataegis, which has been linked previously to APOBEC activity,[35] and ApoHMs were enriched in high-DNA-accessible and highly transcribed regions. This observation reconciles both views, claiming that APOBEC is active during DNA replication (non-clustered and non-hotspot mutations) and in transcriptionally active regions (clustered and hotspot mutations). Additionally, we observed that APOBEC3A-preferred YTCA mutations are dominant in mUC and evenly distributed across genomic regions. This result is in line with experimental observations in human cancer cell lines,[35] suggesting that APOBEC3A is the main driver of

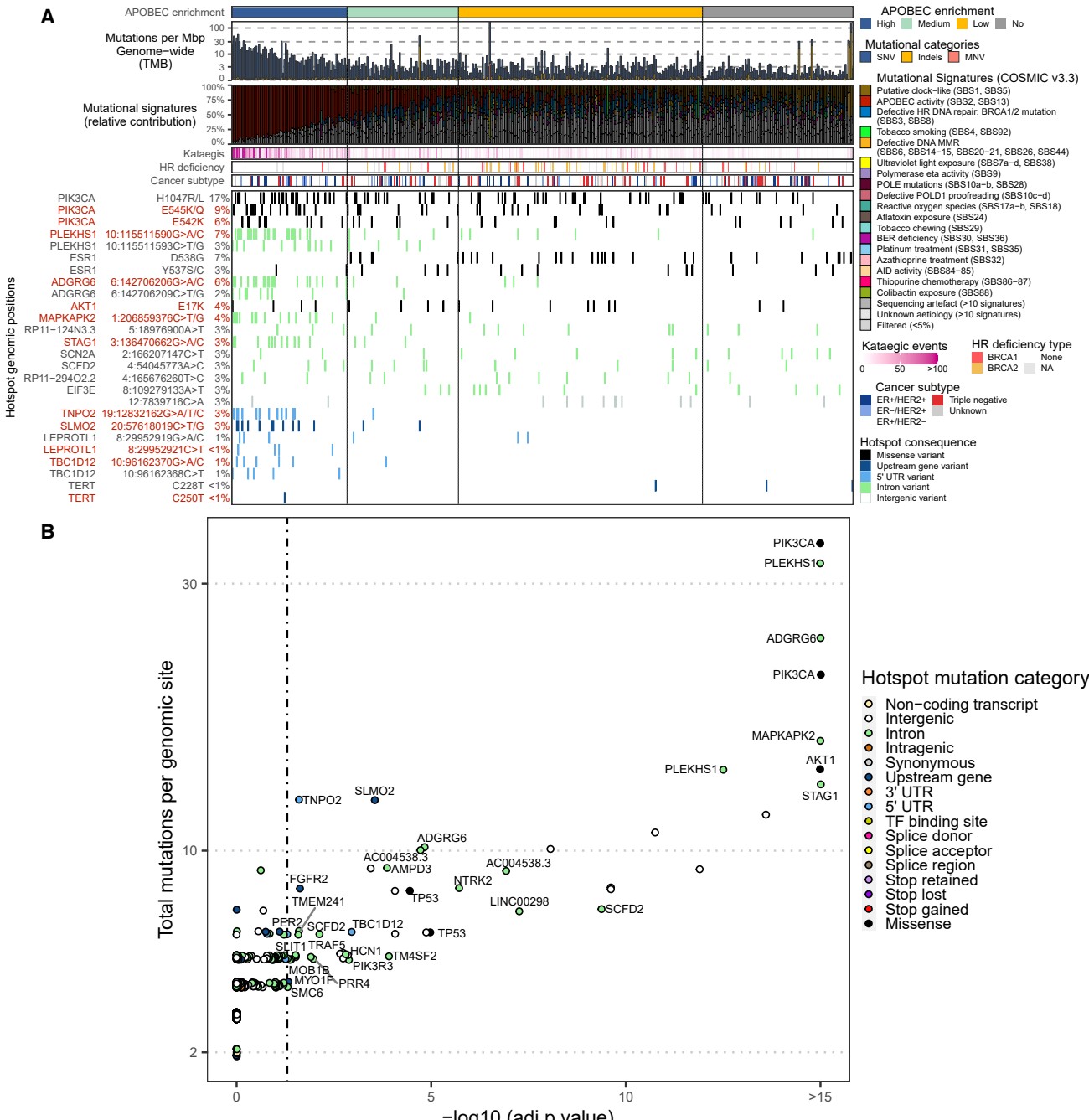

**Figure 6. Genomic landscape and driver hotspot mutations associated with APOBEC mutagenesis in mBC**

(A) WGS data from 442 mBCs were analyzed, and patients were classified according to the enrichment of APOBEC-associated mutations. The genomic features are displayed from top to bottom as follows: APOBEC mutagenesis, genome-wide TMB, COSMIC mutational signatures, frequency of kataegis, HR deficiency and its probable origin, cancer subtype, and the most frequent hotspot mutations (known hotspot mutations in *LEPROTL1*, *TBC1D12*, and *TERT* were also included).

(B) Putative driver hotspot mutations in APOBEC-enriched breast cancer. p-values were adjusted using the Benjamini-Hochberg method (adj p value).

See also Figure S12 and Table S5.

APOBEC mutagenesis. For YTCA mutations as well as for APOBEC3B-preferred RTCA mutations, the gap between the number of mutations across genomic regions is smaller at higher

APOBEC mutagenesis, which strongly suggests that APOBEC enzymes do not have a preference for specific genomic regions. However, the enrichment of highly frequent ApoHMs (n ≥ 4) in

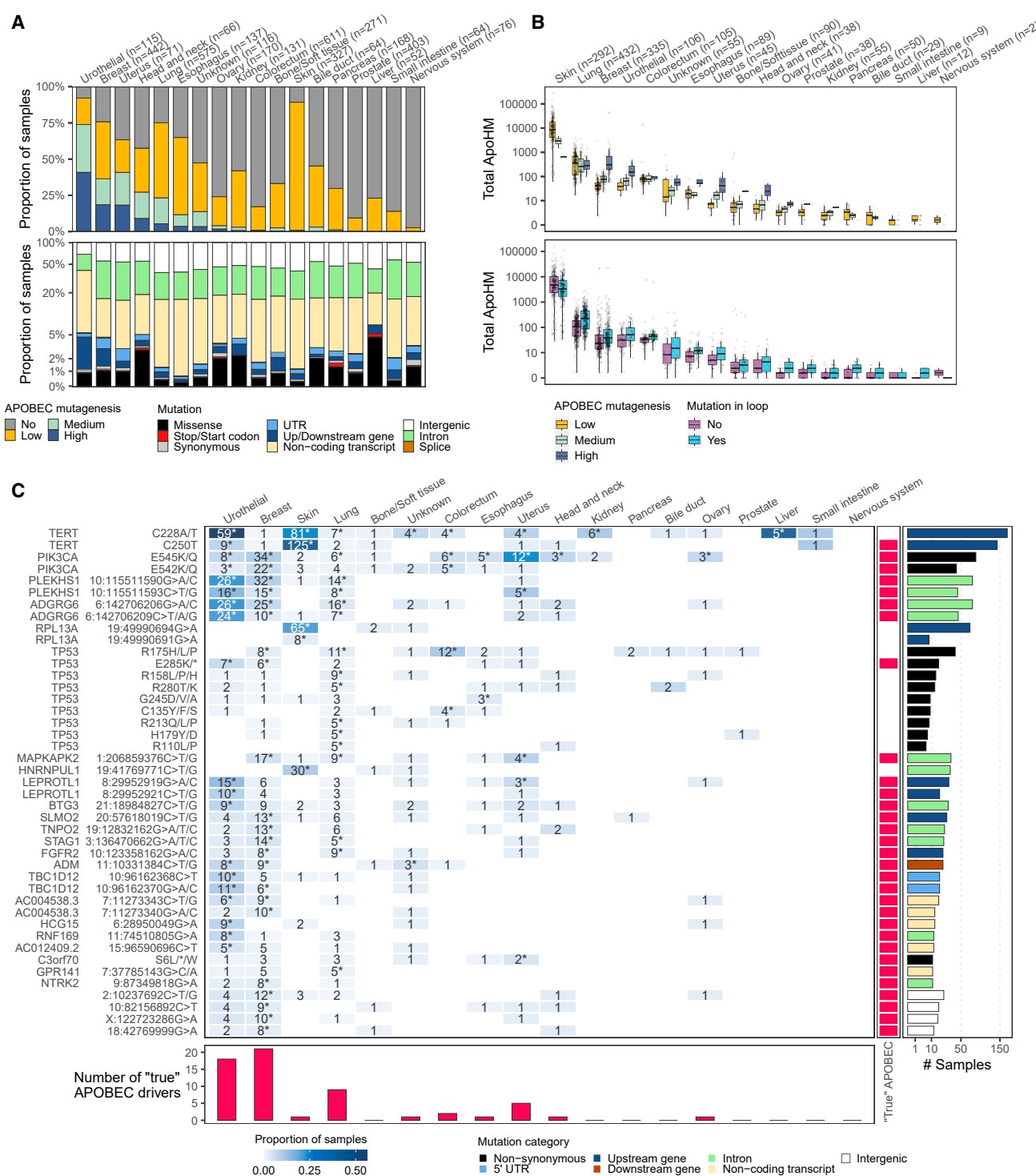

**Figure 7. Pan-cancer overview of ApoHMs and drivers**

(A) Proportion of APOBEC-enriched tumors and hotspot mutations across cancer types.

(B) Distribution of APOBEC-associated hotspot mutations (ApoHMs) across cancer types of APOBEC-enriched tumors. Box plots show the median, inter-quartile range (IQR: Q1–Q3) and whiskers (1.5 × IQR from Q3 to the largest value within this range or 1.5 × IQR from Q1 to the lowest value within this range).

(C) The 10% most frequent genomic positions or genes affected by driver ApoHMs. Drivers of a specific cancer type are indicated by asterisks. The association of driver ApoHMs with APOBEC fold enrichment (continuous values and using all tumor samples) using logistic regression analysis and applying the Wald test (adjusted p < 0.05) shows the "true" APOBEC-derived mutations. All "true" APOBEC-derived driver ApoHMs are included. See also Tables S6 and S7.

open chromatin, of which many are drivers, according to our analysis, may imply a functional effect of these putative driver mutations occurring near gene-regulatory elements.[23]

The extensive examination of ApoHMs in hairpin loops revealed twin mutations we termed didymi, which are characterized by a unique mutational pattern. Didymi comprise the APOBEC SBS2 signature and an unknown signature delineated by C>T mutations in the ApC context. It is remarkable to see only one of the APOBEC signatures in didymi loci when they usually appear together in tumor samples with APOBEC mutagenesis.[36,37] The fact that only C>T mutations that characterize SBS2 are present in didymi suggests that these mutations may arise predominantly by replication across the uracil bases[38,39] and that the mechanisms to generate C>G mutations that characterize SBS13 are not operational in this context. Furthermore, the unknown mutational signature of didymi correlates with spontaneous deamination (SBS1) and defective DNA mismatch repair signatures (SBS6, SBS15, SBS20, and SBS26), suggesting a potentially different mechanism in ApC from the putative TpC APOBEC mutations. Although there is a strong correlation with APOBEC mutagenesis,[22] it is unclear whether both mutations in didymi loci are direct targets of APOBEC3A or whether the non-TpC mutations are just the result of spontaneous deamination followed by DNA mismatch repair.

Compared with breast cancer, twice as many bladder cancer tumors were APOBEC high (19% vs. 41%). Most driver ApoHMs were cancer specific, possibly reflecting different selective pressures that each cancer type endured. In both tumor types, APOBEC-low patients had enrichment for HR deficiency, while APOBEC-high tumors had a high tumor mutational burden, which may indicate different treatment options for these two groups of patients with different levels of APOBEC mutagenesis.[40–43]

The in-depth analysis performed in this study to characterize the mutational landscape of APOBEC mutagenesis revealed the correlation of ApoHMs with the stability of DNA hairpin loops, DNA accessibility, and the potential for didymi associated with these recurrent mutations. These features are key to modeling the background distribution of hotspot mutations and identifying potential drivers. Most potential driver ApoHMs were in the non-protein-coding regions, including didymi. The similar frequency of these drivers in the metastatic and primary settings of UC indicates a general phenomenon in UC, and the drivers could cause early events of tumorigenesis of UC. APOBEC-associated mutations have also been identified in normal tissue;[44,45] however, it is unclear whether the driver ApoHMs we report here are also present in healthy tissue and to what extent they contribute to cancer development from normal cells. Nevertheless, experimental validation will be needed to confirm the cancer driver status of these hotspot mutations.

Although several hotspot mutations are defined as drivers by our model, the inclusion of other cancer types increased the statistical power, revealing that only 31 driver ApoHMs have a strong correlation with APOBECs and may be considered "true" ApoHMs.

In this study, we characterized the genomic landscape of APOBEC-driven mUC and identified novel mechanisms of genomic alteration patterns associated with APOBEC mutagenesis. The mutational signatures associated with DNA hairpin loops targeted by APOBEC in two distinct hotspot positions are unique,

demonstrating the exclusive mutational signature of ApoHMs. These findings were confirmed in non-mUC and in mBC. Additional studies are needed to clarify the role of APOBEC in these recurrent twin mutations. Also, the enrichment of ApoHMs and kataegis in high DNA accessible regions suggests a different mechanism compared with the general APOBEC mutagenesis (non-hotspot mutations) that seems to occur independent of genomic regions, which may be linked to different mechanisms of APOBEC3A and APOBEC3B.[13] As APOBEC is a major source of hotspot mutations, it is crucial to identify those in coding and non-coding regions of whole genomes that may play an important role in cancer development. The statistical framework we developed could aid to identify potential driver hotspot mutations derived from APOBEC activity, which may offer novel targeted therapy options for APOBEC-driven cancer patients.

### Limitations of the study

Despite the thorough analysis we performed, caution should be exercised when considering these outliers as true APOBEC-derived driver hotspot mutations, as other unknown factors may still explain the distribution of these highly frequent APOBEC-related hotspot mutations. The sample size for some tumor types is a limitation when identifying driver hotspot mutations in a cancer-specific manner as we did.

### STAR★METHODS

Detailed methods are provided in the online version of this paper and include the following:

- KEY RESOURCES TABLE
- RESOURCE AVAILABILITY
  ○ Lead contact
  ○ Materials availability
  ○ Data and code availability
- EXPERIMENTAL MODEL AND STUDY PARTICIPANT DETAILS
  ○ Patient cohorts
- METHOD DETAILS
  ○ Whole-genome sequencing and analysis
  ○ Clonal fraction and cancer cell fraction
  ○ Mutational load across genomic regions
  ○ DNA accessibility estimation (ChIPseq)
  ○ Detection of hairpin loops
  ○ Stability of hairpin loops
  ○ Driver hotspot mutations
  ○ RNA-sequencing
  ○ mRNA editing
  ○ Transcriptome expression data mapped to genomic regions
  ○ Simulations and power calculation
- QUANTIFICATION AND STATISTICAL ANALYSIS
  ○ Statistical analysis

### SUPPLEMENTAL INFORMATION

## ACKNOWLEDGMENTS

The Hartwig Medical Foundation and the Center of Personalized Cancer Treatment are acknowledged for making the clinical, genomics, and transcriptomics data available. We would also like to thank the Pan-Cancer Analysis of Whole Genomes study for sharing the genomics data from 23 primary UC patients. We are particularly grateful to all participating patients and their families. The Stichting Dutch Uro-Oncology Study (DUOS) group and the Daniel den Hoed Foundation supported this research. J.A.N.-G. was collectively supported by the National Funding Organization of the Dutch Cancer Society (KWF, the Netherlands) under the framework of the ERA-NET TRANSCAN-2 initiative.

## AUTHOR CONTRIBUTIONS

Conceptualization, J.A.N.-G. and H.J.G.v.d.W.; methodology, J.A.N.-G. and H.J.G.v.d.W.; software, J.A.N.-G., H.J.G.v.d.W., and M.T.W.N.; validation, J.A.N.-G. and H.J.G.v.d.W.; formal analysis, J.A.N. and H.J.G.v.d.W.; investigation, all authors; resources, J.L.B. and J.W.M.M.; data curation, J.A.N.-G., M.R., and H.J.G.v.d.W.; writing – original draft, J.A.N.-G. and H.J.G.v.d.W.; writing – review & editing, all authors; visualization, J.A.N.-G.; supervision, J.L.B., H.J.G.v.d.W., M.P.J.L., and J.W.M.M.; project administration, J.A.N.-G., H.J.G.v.d.W., M.P.J.L., and J.L.B.; funding acquisition, J.L.B., H.J.G.v.d.W., and M.P.J.L.

## DECLARATION OF INTERESTS

J.L.B. has received research support from Merck AG/Pfizer, Janssen, and Merck Sharp & Dohme and consultancy fees from Merck Sharp & Dohme, Bristol-Myers Squibb, Astellas, AstraZeneca, Ipsen, and Janssen (all paid to the Erasmus MC Cancer Institute). M.P.J.L. has received research support from JnJ, Sanofi, Astellas, and MSD and consultancy fees from Incyte, Amgen, JnJ, Bayer, Servier, Roche, INCa, Pfizer, Sanofi, Astellas, AstraZeneca, Merck Sharp & Dohme, Novartis, Julius Clinical, and the Hartwig Medical Foundation (all paid to the Erasmus MC Cancer Institute). J.W.M.M. has received research support from Pfizer, Sanofi, GSK, Therawis Cergentis, and Philips (all paid to the Erasmus MC Cancer Institute) and one consultancy fee from Novartis. A.A.M.v.d.V. has received consultancy fees from BMS, MSD, Pfizer, Novartis, Eisai, Sanofi, Pierre Fabre, Ipsen, and Roche (all paid to the Erasmus MC Cancer Institute).

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

**Cell Genomics**
Article

## STAR★METHODS

### KEY RESOURCES TABLE

| REAGENT or RESOURCE | SOURCE | IDENTIFIER |
|---|---|---|
| **Deposited data** | | |
| Whole-genome DNA- and RNA-sequencing of metastatic cancers | Hartwig Medical Foundation | https://www.hartwigmedicalfoundation.nl/en/data/data-access-request/ |
| Whole-genome DNA-sequencing of non-metastatic urothelial carcinoma | NIH-dbGaP | https://portal.gdc.cancer.gov/ |
| ChIPseq, replication timing and methylation data | The ENCODE Project Consortium and the Roadmap Epigenomics Consortium | https://www.encodeproject.org |
| **Software and algorithms** | | |
| ApobecHM_drivers v1.0.0 | This paper | https://doi.org/10.5281/zenodo.10362579 |
| deepTools v1.30.0 | Ramírez et al.[46] | https://github.com/deeptools/deepTools |
| R v4.1.0 | R Core Team[47] | https://www.r-project.org/ |
| DHARMa v.0.4.6 | Hartig[48] | https://github.com/florianhartig/DHARMa |
| Rediscover v0.3.2 | Ferrer-Bonsoms et al.[49] | https://github.com/cran/Rediscover |
| Genomic subtypes of mUC (GenS1-5) | Nakauma-González et al.[10] | https://bitbucket.org/ccbc/dr31_hmf_muc |
| MutationalPatterns v3.10.0 | Blokzijl et al.[50] | https://bioconductor.org/packages/MutationalPatterns |
| Katdetectr v1.2.0 | Hazelaar et al.[51] | https://bioconductor.org/packages/katdetectr |
| R2CPCT v0.4 | van Riet | https://github.com/J0bbie/R2CPCT |

### RESOURCE AVAILABILITY

#### Lead contact
Further information and requests for resources and reagents should be directed to and will be fulfilled by the lead contact, Harmen J.G. van de Werken (h.vandewerken@erasmusmc.nl).

#### Materials availability
This study did not generate new unique reagents.

#### Data and code availability
- WGS, RNA-seq and clinical data from mUC, mBC and from other metastatic cancers are available through the Hartwig Medical Foundation at https://www.hartwigmedicalfoundation.nl/en/data/data-access-request/, under request numbers HMF: DR-314, DR-026, and DR-041, respectively. For mUC, samples that were previously analyzed (HMF: DR-031) by Nakauma-González et al.[10] were retrieved from HMF: DR-314. WGS data from primary UC was requested to the NCBI dbGAP and granted access through request NIH-dbGaP: #33427.
- ChIPseq, replication timing and methylation data experiments, are freely available through The ENCODE Project Consortium[52] and the Roadmap Epigenomics Consortium[53] on the ENCODE portal (https://www.encodeproject.org).[18]
- The scripts, including the algorithm to find hairpin-loops and estimate the thermodynamic stability have been deposited in a public repository available at https://github.com/ANakauma/ApobecHM_drivers. Additionally, the version v1.0.0 of the code used for this study (ApobecHM_drivers) is available as Data S1 and at Zenodo (https://doi.org/10.5281/zenodo.10362579).[54] Pre-processed WGS data was provided by the Hartwig Medical Foundation and scripts are available at https://github.com/hartwigmedical/hmftools. R2CPCT v0.4 was used for additional processing of the WGS (https://github.com/J0bbie/R2CPCT),

### EXPERIMENTAL MODEL AND STUDY PARTICIPANT DETAILS

#### Patient cohorts
The mUC cohort of this study has been previously described (NCT01855477 and NCT02925234).[10] In short, patients with advanced or mUC were prospectively enrolled in these multicenter clinical trials and were scheduled for 1st or 2nd line palliative systemic treatment. Following protocols of the Hartwig Medical Foundation (HMF),[55] WGS, with a depth close to 100X[10], was successfully

performed on DNA from freshly obtained biopsies from metastatic sites, and matched RNA-sequencing (RNA-seq) was available for 90 patients (97 samples). Sequential biopsies of a metastatic lesion taken at the time of clinical or radiological disease progression from eight patients were additionally sequenced. Similarly, the cohorts of other cancer types have been previously described,[30] and the DNA extraction and sequencing were performed according to the HMF protocols.[55] Only cancer types with >50 samples were included in the analysis.

## METHOD DETAILS

### Whole-genome sequencing and analysis

Alignment and pre-processing of WGS data, detection of genomic subtypes, HR deficiency, MSI, structural variants, chromothripsis events, APOBEC mutagenesis, and pathway activity have been previously described.[4,10,55,56] Mutational signatures and kataegis were detected with MutationalPatterns v3.10.0[50] and Katdetectr v1.2.0.[51] APOBEC enriched tumors (adj. p < 0.01, otherwise non-APOBEC tumors) were classified as high when the fold enrichment ($E$) for C>T and C>G mutations in T$\underline{C}$W (W = A or T) context was $E \geq 3$, medium when $2 \leq E < 3$ and low when $E < 2$. Similarly, the fold enrichment for C>T and C>G mutations in the tetra YT$\underline{C}$A (Y = T or C) and RT$\underline{C}$A (R = G or A) context was calculated.

### Clonal fraction and cancer cell fraction

The clonal fraction of mutations was estimated as previously described.[57] Correcting for tumor purity and copy number, the variant copy number $n_{SNV}$ of each SNV was calculated as follows

$$n_{SNV} = \frac{f_m}{p}[pC_t + (1 - p)C_h],$$ (Equation 1)

where $f_m$ is the relative frequency of the mutant variant reads, $p$ is the tumor purity, $C_t$ is the copy number affecting the region where a particular SNV was located and $C_h$ is the healthy copy number (2 for autosomes and 1 for allosomes). In this study, mutations were considered clonal when the variant copy number was >0.75.

To identify the proportion of cancer cells carrying a specific mutation, the cancer cell fraction (CCF) was estimated as previously described.[58,59] Given the number of reference and mutant reads and assuming binomial distribution, we estimated the expected number of allelic copies ($n_{chr}$) carrying the observed SNV resulting from $f_m$ values when the mutation is present in 1, 2, 3, …, $N_{chr}$ allelic copies. The resulting estimated $n_{chr}$ with the maximum likelihood is used to calculate the CCF as $n_{SNV}/n_{chr}$.

### Mutational load across genomic regions

The genome was divided into regions (bins) of one mega base-pair (Mbp). The number of SNVs was counted in each bin, and the mean number of SNVs was estimated from the entire cohort. These values represented the average SNVs/Mbp reflecting the mutational load in each genomic region. The average SNVs/Mbp was smoothed by applying a moving average with a window of k = 3. For visualization reasons, in Figure 2 a k = 9 was used.

### DNA accessibility estimation (ChIPseq)

ChIPseq data for healthy urinary bladder, breast and other tissues of adult humans (H3K4me1, H3K4me3, H3K36me3 and H3K27ac) were downloaded from the ENCODE portal (https://www.encodeproject.org) to our local server. The bed.gz files were imported with narrowPeak format for analysis. Only peaks with q < 0.05 were kept for analysis. The signal of each experiment was divided into regions of one Mbp, and a moving average with k = 3 bins was applied. The signals were normalized using the mean and standard deviation. This procedure was applied to each chromosome. The sum of all four ChIPseq experiments was considered an approximation of DNA accessibility. High DNA accessible regions (open chromatin) had values above the median considering the whole genome. All other regions were considered to be of low DNA accessibility (condensed chromatin). DNA accessibility for all healthy tissues is available in Table S7 (for urinary bladder see Table S4 along with other covariates). In case that matched normal ChIPseq with tumor type was not available, an average of all ChIPseq experiments was used.

### Detection of hairpin loops

All SNVs were assessed to determine whether they occur in the loop of hairpin-loop structures and their thermodynamic stability. A total of 50 bases upstream and downstream of the mutation site were considered. The minimum length of the stem was 2 base-pairs and the minimum and maximum loop size was 3 and 10 bases, respectively (not considering the closing base-pair). Since multiple configurations are possible, only the structure with the highest stability was considered. One mismatch was allowed which could be either a non-matching base-pair or a single nucleotide bulge loop.

### Stability of hairpin loops

We implemented the nearest neighbor stability algorithm (NNSA) to estimate the thermodynamic stability of hairpin-loops. This algorithm calculates the Gibbs free energy ($\Delta G$) of the DNA hairpin-loop structure based on known biophysics properties of the base pairs and their interactions.[60,61]

The NNSA was applied to the target DNA sequence allowing mismatches in the stem. Each base-pair contributes to the stability of the stem considering the immediate neighboring base-pairs. Adding the local $\Delta G$'s calculated per base and accounting for the size and sequence of the loop results in a final $\Delta G$ for the whole hairpin-loop structure. All parameters are available in the literature and the UNAFold web server was used to infer parameters for mismatches.[60–62] For loops larger than 4, no data was available for specific sequences and only the loop size was considered.

### Driver hotspot mutations

To identify driver hotspot mutations, all genomic positions with 2 or more mutations were considered for analysis. Hotspot mutations were divided into two groups either located within loops of hairpin-loops or outside of these DNA structures. For hotspot mutations outside of loops, only those in the TpC context were considered, as these are likely initiated by APOBEC enzymes. For hotspot mutations falling within loops, all hotspot mutations in NpC (N = any base) context were considered, as APOBEC3A may also mutate these bases that are not in the TpC context.[22]

In cancer, only a few somatic mutations are drivers, while the vast majority are passenger mutations.[31] Under this consideration, we modeled the distribution of the remaining hotspot mutations as a Poisson process per focal hotspot mutation. For more accurate modeling, we considered the tri-nucleotide context TCW (TCA, TCC, TCG, TCT). In the case of hotspot mutations that do not occur in hairpin loops, DNA accessibility was used as a predictor variable that can influence the distribution of hotspot mutations. We modeled this in R as model_TCW_noloop = glm(n_mut_genpos+DNA_access). Where, n_muts_genpos is a vector with the number of mutations per genomic position linked to the DNA accessibility region (DNA_access). Accessibility regions were divided into 10 regions based on percentiles. Since DNA-accessibility varies per tumor-tissue of origin,[63,64] this was estimated for mUC, mBC and other tumor types using ChIPseq experiments from normal tissue as described above.

In the case of hotspot mutations within hairpin-loops, the model was extended to include mutations in non-TCW context (grouped as ApC, CpC or GpC context), the hairpin loop stability (hairpin_stab) and the DNA sequence in the loop (loopSeq): model_TCW_loop = n_mut_genpos+DNA_access+hairpin_stab+loopSeq. The loop sequence was a binary variable to indicate whether the mutation occurred in the following sequence: $\underline{1}$001 or $\underline{1}$01 (0 = A/T, 1 = G/C; underlined is the position of the hotspot mutation).

Using these models, we estimated the expected number of mutations of the specified hotspot mutation for which the model was built. Then, the exact Poisson test was applied to estimate the significance of observing the same or more mutations than expected in a specific genomic position. p-values were Benjamini-Hochberg adjusted. In rare occasions, only a few mutations (<2) were available to represent the background distribution of a particular tri-nucleotide. To include these ApoHM in the analysis, a model that did not consider the tri-nucleotide was used instead.

### RNA-sequencing

Alignment, pre-processing of RNA-seq data and transcript normalization have been previously described.[10,30] The transcriptomic subtype of each mUC sample was identified when the mean (normalized) expression of all genes associated with a specific subtype was the highest across all subtypes.

### mRNA editing

Jalili et al. identified hotspot mutations in the mRNA of *DDOST* that is targeted by APOBEC3A.[17] The genomic position of this hotspot mutation reveals a hairpin-loop structure that is an ideal substrate for APOBEC3A. Due to the short life-time of mRNA molecules, the presence of this hotspot mutation reflects ongoing APOBEC mutagenesis. The proportion of C > U mutations in chr1:20981977 was estimated to identify the RNA-editing activity of APOEBC3A.

### Transcriptome expression data mapped to genomic regions

MultiBamSummary from deepTools v1.30.0[46] was used to read BAM files and estimate the number of reads in genomic regions with a size of one Mbp. The average raw read count per Mbp was calculated, and a moving average with k = 3 bins was applied. The scale of the read counts was normalized per chromosome using the mean and standard deviation. High transcriptional regions were defined as such when the expression value of one region was above the median of the whole genome.

### Simulations and power calculation

A synthetic genome with 1,000,000 hotspot mutations was reconstructed from the original cohort of mUC. To reach the number of hotspot mutations, non-hotspot mutations were randomly selected and the number of mutations per genomic position was drawn from a Poisson distribution using the empirical lambda from the mUC cohort. The same number of driver ApoHM identified in mUC were simulated as hypothetical drivers to replicate a 3%–15% prevalence. Hypothetical cohorts with 10–500 samples were simulated 100 times, using a random number of ApoHM derived from the empirical distribution of the mUC cohort. The statistical power was estimated as the proportion of driver ApoHM that were correctly identified. The performance of the model on simulated cohorts of 100 samples, was also tested with other genomic covariates. These covariates were replication timing and methylation from HeLa cell lines, and the proportion of GC content from the reference build hg19.

## QUANTIFICATION AND STATISTICAL ANALYSIS

### Statistical analysis

Analyses were performed using the statistical analysis platform R v4.1.0.[47] Fisher's exact, Wilcoxon-rank sum and Wilcoxon signed-rank tests were used for comparison between groups. The correlation coefficients of continuous values with categorical values were estimated with logistic regression analysis applying the Wald test. Residuals for QQ plots and the Kolmogorov-Smirnov test were estimated using DHARMa v.0.4.6.[48] The exact Poisson test was applied to identify potential driver hotspot mutations. The Poisson-binomial method was applied for mutually exclusive mutation events using Rediscover v0.3.2[49] and the Fisher's exact test was applied for the significance of co-occurred mutations. In all cases, p values were adjusted using the Benjamini-Hochberg method.

