## [Document S2. Transparent peer review records for Nakauma-González et al · Cell Genomics]

Whole-genome mapping of APOBEC mutagenesis in metastatic urothelial carcinoma identifies driver hotspot mutations and a novel mutational signature

J. Alberto Nakauma-González, Maud Rijnders, Minouk T. W. Noordsij, John W. M. Martens, Astrid A.M. van der Veldt, Martijn P. J. Lolkema, Joost L. Boormans, Harmen J. G. van de Werken

Summary

Initial submission: Received : Sep 15, 2023

Scientific editor: Sara Rohban

First round of review: Number of reviewers: 2
Revision invited : Oct 16, 2023
Revision received : Dec 22, 2023

Second round of review: Number of reviewers: 2
Accepted : Mar 06, 2024

Data freely available: YES

Code freely available: YES

This transparent peer review record is not systematically proofread, type-set, or edited. Special characters, formatting, and equations may fail to render properly. Standard procedural text within the editor's letters has been deleted for the sake of brevity, but all official correspondence specific to the manuscript has been preserved.

Referees' reports, first round of review

Reviewer #1:

In this manuscript, authors utilized 115 WGS data from the metastatic urothelial cancer (mUC) cohort to characterize the APOBEC mutagenic process and identify APOBEC hotspot drivers. Overall, they found APOBEC hotspot mutations to be enriched in open chromatin regions. Similarly, the found DNA sequence forming hairpin loops frequently consisted of "didymi" (two APOBEC-associated hotspot mutations). Finally, they build a statistical model to identify cancer drivers in mUC and metastatic breast cancer.

The study taking a deep dive into APOBEC mutagenesis is very interesting and timely. However, there are some clarifications and concerns that authors should address, which are listed below.

1. Can this study be extended to other metastatic or primary cancer types? Can authors comment on the sample size and statistical power to uncover ApoHM drivers for the mUC cohort presented in this study?
2. Authors noted the presence of subclonal populations due to APOBEC mutagenesis in the evolution of UC metastasis. Can they characterize this by looking at APOBEC mutation distribution based on the subclonal status of mutations? How does this observation compare between the primary vs metastatic cancer cohort?
3. Why recurrence frequency threshold of ApoHM was chosen at the value of 4?
4. It was not entirely clear why ApoHM didn't influence gene expression but were associated with pathway related to transcriptomic effects.
5. In Figure 3a, what does the downstream gene variant represent? Also, what fraction of recurrent APOBEC mutations fall in different cis-regulatory elements?
6. How is hairpin loop stability related to APOBEC3A mutagenesis? They use stability as a feature for their statistical model for driver prediction. However, it will be good to get a general insight into this by surveying APOBEC mutation distribution in stable vs unstable hairpin loops.
7. It's unclear whether the authors used normal or cancerous cell lines for the breast and bladder in the current study. The authors should clarify this.
8. Why won't other genomic covariates (methylation, replication timing, GC content, etc) be relevant for their statistical model for identifying APOBEC drivers?
9. How does patient survival correlates among APOBEC-high and APOBEC-low patients?

Reviewer #2:

The authors present an analysis of APOBEC mutations in metastatic urothelial cancers, a tumor type where this process appears to be particularly active. The analysis is generally well-performed, the text is clear and concise and the findings are interesting. Many of the observations are confirmatory - APOBEC mutagenesis in human cancers has been well studied over the last decade and many of the basic conclusions here have been previously described. Where the manuscript has the most novelty is in the analysis of APOBEC hotspots, particularly around palindromic / hairpin sequences in DNA. They describe this interesting subset of variants in considerable detail, and this is interesting.

The analysis that I was least convinced about was the analysis of whether these hairpin sequences could act as driver mutations. This analysis is purely genetic, based on a model to predict the expected rate of mutagenesis depending on the properties of the hairpin. Several hotspots maintained a higher rate than predicted by this background model, which the authors interpret as evidence of selection acting on the variants. This is a challenging area. Most algorithms for the inference of selection benefit from a broadly even, random distribution of mutations across the genome, with the various covariates (gene expression, sequence context, replication timing etc) causing limited variation in this rate. However, in the setting addressed here, the variability in mutation rates across these hairpins represents several orders of magnitude, and my prediction is that the covariates only capture a fraction of this variability. The authors should present much more detail about the accuracy of predictions (R^2 ; QQ plots; leave-one-out cross-validation etc) in their model. I would also recommend that more caution is deployed in the text - either the outliers are drivers (as argued by the authors) or there are variables, yet undiscovered, which influence the rate of APOBEC mutagenesis at these hotspots (the explanation I favor).

Authors' response to the first round of review**Reviewer #1**

In this manuscript, authors utilized 115 WGS data from the metastatic urothelial cancer (mUC) cohort to characterize the APOBEC mutagenic process and identify APOBEC hotspot drivers. Overall, they found APOBEC hotspot mutations to be enriched in open chromatin regions. Similarly, they found DNA sequence forming hairpin loops frequently consisted of "didymi" (two APOBEC-associated hotspot mutations). Finally, they build a statistical model to identify

cancer drivers in mUC and metastatic breast cancer. The study taking a deep dive into APOBEC mutagenesis is very interesting and timely. However, there are some clarifications and concerns that authors should address, which are listed below.

1. Can this study be extended to other metastatic or primary cancer types? Can authors comment on the sample size and statistical power to uncover ApoHM drivers for the mUC cohort presented in this study?

Response:

*We thank R#1 for assessing our study. We have extended the analysis to include 16 other cancer types using whole-genomes of 3302 metastatic samples (+115 mUC +442 mBC). We have summarized the findings in the main Figure 7 and lines 276-299 of the main text. In short, we found that urothelial, breast, and uterus metastatic cancers have the highest proportion of APOBEC-high tumors (**Figure 7A**) and that the number of ApoHM increases with the strength of APOBEC mutagenesis (**Figure 7B**). This larger dataset of $n=3859$ samples allowed us to identify driver hotspot mutations with a strong positive correlation with APOBEC enrichment. This suggests that the “true” APOBEC-derived driver hotspot mutations are only 31 (**Figure 7C**). We have also included a power calculation based on simulations of a synthetic genome of mUC. We show that to reach a power of 80%, we will need a cohort of 400 samples. However, for highly frequent hotspot mutations (>10% in a cohort), a cohort of ~75 samples is sufficient. This finding together with the performance of the model is presented in **Figure S11**.*

2. Authors noted the presence of subclonal populations due to APOBEC mutagenesis in the evolution of UC metastasis. Can they characterize this by looking at APOBEC mutation distribution based on the subclonal status of mutations? How does this observation compare between the primary vs metastatic cancer cohort?

Response:

*We have interrogated the mutational signatures of mutations based on the subclonal status. We found that APOBEC is present in clonal and non-clonal mutations of metastatic UC (**Figure S5**). The subclonal status of mutations in primary tumors was not possible to determine due to the lack of the necessary information to correct for purity and copy number. However, we used the variant allele frequency and found similar results as in the metastatic cohort (**Figure S5B**).*

3. Why recurrence frequency threshold of ApoHM was chosen at the value of 4?

Response:

Hotspot mutations above 4 are rare in the mUC cohort, thus we decided to group them (as very frequent hotspots) for visualization purposes only. For statistical analysis, all hotspot mutations (≥ 2 mutations per genomic position) were considered.

4. It was not entirely clear why ApoHM didn't influence gene expression but were associated with pathway related to transcriptomic effects.

Response:

We reported no correlation of ApoHM with expression of the affected gene. For some hotspot mutations, such as those affecting FGFR3, there is a clear correlation with FGFR3 expression. For other hotspot mutations, such as in TERT, there is no association with the expression of TERT. It is unclear to us, why this is the case, however, tumors carrying these hotspot mutations show a distinct transcriptomic profile, and a different transcriptional state is often associated with a different phenotype (PMID: 33033407). The direct influence of these hotspot mutations with changes in the transcriptome cannot be established, but only associations can be reported. To avoid confusion, we have updated the text to exclude any reference linking hotspot mutations and phenotypes.

5. In Figure 3a, what does the downstream gene variant represent? Also, what fraction of recurrent APOBEC mutations fall in different cis-regulatory elements?

Response:

We used VEP annotations and according to the definition, downstream gene variants are non-coding variants located within 5,000 bases 3' of a gene. This default distance was used in the WGS processing pipeline.

*The summary of APOBEC hotspot mutations across the genome is presented in **Figure 5A**. The distribution of ApoHM according to their frequency (left column/panel) and hairpin stability (middle panel) shows that most mutations occur in non-coding transcripts, intergenic or intronic regions. This pattern is also observed in all hotspot mutations across different cancer types (**new Figure 7A**), where ~95% of all hotspot mutations occur in these genomic regions. However, this varies per cancer type and can represent up to 99% of all hotspot mutations in esophagus cancer. Similarly, the proportion of hotspot mutations in different regulatory elements may vary per cancer type.*

6. How is hairpin loop stability related to APOBEC3A mutagenesis? They use stability as a feature for their statistical model for driver prediction. However, it will be good to get a general insight into this by surveying APOBEC mutation distribution in stable vs unstable hairpin loops.

Response:

The link between APOBEC3A and mutations in hairpin loops has been shown in cancer and validated in experimental settings, which we cite in the main text (line 215). We confirmed this link and showed it in Figure 7A, as stated in lines 215-216. Because this has been analyzed in detail by others, we did not extensively explain it in the text, but what we see in the center of

*the lower panel is that low frequent APOBEC hotspot mutations ($n = 2$) are dominated by unstable hairpin loops, while very frequent mutations have a more stable hairpin loop. For this reason, we use the stability of hairpin loops as a covariate in our model. We also showed the added value of hairpin loops in the model (**Figure S11D**; see also response to the comment of R#2).*

7. It's unclear whether the authors used normal or cancerous cell lines for the breast and bladder in the current study. The authors should clarify this.

Response:

To assess our model in other cancer types, we used a cohort of metastatic breast cancer. To infer methylation levels and replication timing, we used results from cell lines. This information has been added to the methods. The contribution of these genomic features to the model has been analyzed as described in the following question.

8. Why won't other genomic covariates (methylation, replication timing, GC content, etc) be relevant for their statistical model for identifying APOBEC drivers?

Response:

*We have tested other genomic covariates (GC content, RNA expression levels, mutational load, methylation and replication timing) and found that the baseline model that only considers the trinucleotide context can identify many drivers that are also identified by models that consider different genomic covariates. However, the baseline model also has a high number of false positives (not found in other models). The addition of hairpin loop information and ChIP-seq data increases the R-squared of the model (see response to comment of R#2) and the number of driver hotspot mutations identified while reducing false positives. Due to the high (anti-)correlation between the different genomic features (**Figure S11C**), using other genomic covariates in the model instead of ChIP-seq data gives similar results. Interestingly, combining multiple genomic covariates increases the average number of driver hotspot mutations, but at the cost of more false positives. This comment is in line with R#2 and the results are presented in **Figure S11D** and summarized in the main text lines 240-254.*

9. How does patient survival correlates among APOBEC-high and APOBEC-low patients?

Response:

APOBEC-high tumors are associated with a better prognosis as reported in the primary UC TCGA cohort (PMC5687509). We also recently reported that APOBEC-high tumors (using the same definition as in this study) have a longer survival probability than other tumors when treated with immunotherapy. We have mentioned this in the text with the corresponding references in line 51.

Reviewer #2:

The authors present an analysis of APOBEC mutations in metastatic urothelial cancers, a tumor type where this process appears to be particularly active. The analysis is generally well-performed, the text is clear and concise and the findings are interesting. Many of the observations are confirmatory - APOBEC mutagenesis in human cancers has been well studied over the last decade and many of the basic conclusions here have been previously described. Where the manuscript has the most novelty is in the analysis of APOBEC hotspots, particularly around palindromic / hairpin sequences in DNA. They describe this interesting subset of variants in considerable detail, and this is interesting.

The analysis that I was least convinced about was the analysis of whether these hairpin sequences could act as driver mutations. This analysis is purely genetic, based on a model to predict the expected rate of mutagenesis depending on the properties of the hairpin. Several hotspots maintained a higher rate than predicted by this background model, which the authors interpret as evidence of selection acting on the variants. This is a challenging area. Most algorithms for the inference of selection benefit from a broadly even, random distribution of mutations across the genome, with the various covariates (gene expression, sequence context, replication timing etc) causing limited variation in this rate. However, in the setting addressed here, the variability in mutation rates across these hairpins represents several orders of magnitude, and my prediction is that the covariates only capture a fraction of this variability. The authors should present much more detail about the accuracy of predictions (R^2 ; QQ plots; leave-one-out crossvalidation etc) in their model. I would also recommend that more caution is deployed in the text - either the outliers are drivers (as argued by the authors) or there are variables, yet undiscovered, which influence the rate of APOBEC mutagenesis at these hotspots (the explanation I favor).

Response:

*We thank R#2 for reading and commenting on our study. We have tested the performance of our model by considering different scenarios, which are now depicted in **Figure S11**. First, we calculated the McFadden's R-squared and saw a slight increase in the goodness of fit when considering the hairpin loop ($R^2 = 0.23$), sequence in the loop ($R^2 = 0.27$) and DNA accessibility regions ($R^2 = 0.28$). Second, we evaluated the contribution of the covariates to identify outliers. As already shown by the R^2 , the contribution of genomic features only captures a small fraction of the variability which translates into a small increase of drivers identified. We also tested other covariates that are often used to model the mutational background in cancer and reached similar conclusions. Additionally, only one of the genomic features is necessary as they all show a high correlation, making them redundant. Third, the Q-Q plots show that the empirical distribution of ApoHM deviates from the theoretically expected distribution (Kolmogorov-Smirnov test $p < 0.001$). However, when outliers that represent highly frequent ApoHM (>10)*

are excluded, which according to our analysis are all drivers, we observed a good agreement between our model with the theoretical distribution (**Figure S11A**; Kolmogorov–Smirnov test $p = 0.19$). An overview of the performance of the model is described in the main text lines 240-254.

Finally, we agree that other unknown factors may still explain the distribution of these highly frequent APOBEC-related hotspot mutations. Thus, we have updated the discussion to include a statement in lines 355-360 regarding the caution that needs to be exercised when considering these outliers as true driver hotspot mutations.

Referees' report, second round of review

Reviewer #1:

Authors have done several new analyses and reported them in the revised manuscript. I have no additional concerns.

Reviewer #2:

The authors have responded well to my comments. I have no further suggestions for improvement.

Authors' response to the second round of review

NA